# A Biochemical and Structural Understanding of TOM Complex Interactions and Implications for Human Health and Disease

**DOI:** 10.3390/cells10051164

**Published:** 2021-05-11

**Authors:** Ashley S. Pitt, Susan K. Buchanan

**Affiliations:** Laboratory of Molecular Biology, National Institute of Diabetes and Digestive and Kidney Diseases, National Institutes of Health, Bethesda, MD 20892, USA; Susan.Buchanan2@nih.gov

**Keywords:** TOM complex, mitochondrial quality control, mitochondrial cell signaling, TOM subunits, TOM complex interactions

## Abstract

The central role mitochondria play in cellular homeostasis has made its study critical to our understanding of various aspects of human health and disease. Mitochondria rely on the translocase of the outer membrane (TOM) complex for the bulk of mitochondrial protein import. In addition to its role as the major entry point for mitochondrial proteins, the TOM complex serves as an entry pathway for viral proteins. TOM complex subunits also participate in a host of interactions that have been studied extensively for their function in neurodegenerative diseases, cardiovascular diseases, innate immunity, cancer, metabolism, mitophagy and autophagy. Recent advances in our structural understanding of the TOM complex and the protein import machinery of the outer mitochondrial membrane have made structure-based therapeutics targeting outer mitochondrial membrane proteins during mitochondrial dysfunction an exciting prospect. Here, we describe advances in understanding the TOM complex, the interactome of the TOM complex subunits, the implications for the development of therapeutics, and our understanding of the structure/function relationship between components of the TOM complex and mitochondrial homeostasis.

## 1. Introduction

Mitochondria are double membrane-bound organelles primarily responsible for energy production in eukaryotes. In addition to the primary function of energy production, mitochondria also play roles in other essential cellular processes, namely lipid synthesis, calcium homeostasis, apoptosis, cell differentiation, and signaling [1]. The importance of mitochondria to cellular function and homeostasis makes the organelle an essential target for understanding human health and disease.

The heavy reliance of cardiac tissue, skeletal muscle, the brain, and the liver on the cellular functions of mitochondria makes the study of mitochondrial biology essential for understanding organ function, how mitochondrial dysfunction contributes to diseases that impact these organ systems, and understanding of mitochondrial diseases [2]. The study of mitochondrial dysfunction has contributed to our understanding of a broad set of diseases such as neurodegenerative diseases, cancer, and metabolic diseases [3].

The mitochondrial entire proteome is about 1200 proteins in yeast, and 1500 in humans, but the bulk of it is nuclear-encoded and imported from the cytoplasm, despite the presence of a mitochondrial genome and ribosome in the organelle [4,5]. Proteins destined for the mitochondrion have locations in one of several mitochondrial compartments: the outer mitochondrial membrane (OMM), the inner mitochondrial membrane (IMM), the intermembrane space (IMS), or the matrix. Accurate targeting and sorting of proteins require a host of protein import machinery, sorting machinery, and chaperones to aid in folding and insertion of proteins (Figure 1). In humans, just 13 proteins are encoded and translated in mitochondria and all are inner mitochondrial membrane proteins. In addition to encoding for proteins, mitochondrial DNA plays a critical role in the innate immune response [6].

The primary entry point for nuclear-encoded preproteins into mitochondria is the translocase of the outer membrane (TOM) complex. The channel-forming component of the TOM complex is a 19 stranded β-barrel protein, Tom40. Across eukaryotes, Tom40 is conserved and is even present in *Giardia intestinalis* (which lack mitochondria but contain mitosomes), a mitochondrion-related organelle lacking genes required for oxidative phosphorylation [7]. Studies in *Saccharomyces cerevisiae* and *Neurospora crassa* have demonstrated that Tom40 is essential for cell viability [8]. Tom40 works in concert with other subunits that demonstrate roles in translocation, biogenesis of the TOM complex, and regulating the dynamics of the mature complex. The canonical TOM complex identified in *S. cerevisiae* and *N. crassa* has seven core subunits, Tom40, Tom22, Tom5, Tom6, Tom7, and two loosely associated receptor subunits Tom20 and Tom70. The components of the TOM complex show slight variation amongst different species. However, in most eukaryotes, Tom40, Tom22, and Tom7 are the most conserved. Most studies on the TOM complex have focused on the fungal homologs; however, some studies in humans show that various subunits play specific roles in neurodegeneration, innate immunity, hepatic regeneration, and cellular stress response.

The elaborate network of proteins responsible for maintaining the mitochondrial proteome also includes interactions with proteins that play roles in protein degradation, metabolite transport, and post-translational modifications. As the general import pathway for mitochondrial proteins, in addition to interacting with proteins during translocation, TOM complex subunits also interact with mature proteins to carry out mitochondrial functions. Some crosstalk between components of the mitochondrial import machinery serves import functions; however, there is still much uncertainty regarding the other functions these interactions may have.

In this review, we discuss advances in our understanding of the mitochondrial import machinery, of the interactome of TOM subunits as it relates to human health and disease, and of mitochondrial protein import dysregulation that teaches us about pathology and the development of therapeutics (Figure 2).

## 2. Structure and Functional Data on the TOM Complex

### 2.1. Biochemical and Structural Characterization of the Components of the TOM Complex

The TOM core complex is comprised of components Tom40, Tom22, and the β-barrel associated subunits Tom5, Tom6, Tom7 (Figure 3A). The structures of each component have only been determined in the context of the TOM core complex and very little structural information is available on the conformations of these subunits in complex with other proteins or free in the membrane. Biochemical studies in yeast have illuminated much of what we know about the functions of each of these subunits. Tom40 is the main import pathway of preproteins and serves as the major structural feature of the TOM complex. All current structures of the TOM complex show that the Tom40 β-barrel contains an N-terminal α-helix, and in the structures of the human TOM (hTOM) complex, Tom40 measures 30 Å by 40 Å excluding this N-terminal α-helix [9] (Figure 4B).

The N-terminal helix that extends inside the β-barrel has an interaction interface that is highly conserved across species, although mammalian N-terminal domains are extended compared to other eukaryotic homologs. Experiments focused on elucidating the role of the hTom40 N-terminal α-helix demonstrate that the fusion of GFP to this helix prevents proper targeting of hTom40 to the mitochondria, which was not observed with C-terminal fusion proteins, indicating that there is some importance to the N-terminus for targeting. Interestingly, hTom40^D2-44^, a mutant with a portion of this region deleted, is able to insert into mature TOM complexes and has similar stable biogenesis intermediates compared to full-length hTom40 [10]. N-terminal truncations of hTom40 revealed that deletion of the conserved residues 72–87 results in arrest of hTom40 precursors at the human sorting and assembly machinery (hSAM) complex. The hSAM complex, like its fungal homolog, is responsible for the integration of β-barrels such as Tom40, VDAC, and the β-barrel in the complex itself, Sam50 [11]. Interestingly, some of these residues, notably N79, P80, T82, and H87 are shown in the structure to hydrogen bond to the inner lumen wall of the β-barrel which suggests a role in stabilizing the barrel or perhaps plays an active role in gating protein import. The import and integration of hTom40 into the mature TOM complex has distinct stable intermediates from those observed in fungi, suggesting differences in biogenesis mechanism in humans from their fungal counterparts [10].

In the reported structure of hTom40, density is missing for residues 1–76, and the visible remaining portion of this conserved region forms a coil followed by an α-helix (Figure 4C). Further structural studies targeting hTom40 intermediates are necessary to determine the structural role that this region plays in the biogenesis of the hTOM complex. Available structural knowledge, along with gene interaction studies, provides information on how hTom40 mutations may contribute to disease states [12].

Receptor proteins that associate with Tom40 are important in recognition of mitochondrial targeting sequences and in ensuring the fidelity of mitochondrial protein import. The core receptor protein of the human TOM complex is the central receptor, Tom22, which serves as a major site for preprotein binding and plays an organizational role in the assembly of the TOM complex. Knockdown of hTom22 results in increased amounts of stable hTOM complex intermediates, which agrees with reports that hTom40 assembly relies on hTom22. Overexpression of Tom22 suggests that hTom22 is rate-limiting for the assembly of complete TOM complexes and promotes the integration of small β-barrel-associated subunit hTom7 in HeLa cells. Most recent structures of the hTOM complex reveal that Tom22 sits between the two Tom40 β-barrels and serves as the dimeric interface as well as serving a stabilizing role for Tom40 [9]. Tom22, a single spanning amphipathic α-helical protein, contains some interesting structural features, most notably a kink caused by a conserved proline (Figure 5A). Differences between the reported fungal structures and the human structure of Tom22 include a relatively vertical insertion of the transmembrane segment as opposed to the lateral tilt seen in the fungal Tom22 structures. Early studies on the biochemical elements necessary for import in fungi note the presence of acidic patches on the cytosolic domains of Tom20, Tom22, Tom5, and the IMS domains of Tom7 and Tom22 [13,14,15]. These acidic patches were hypothesized to be essential for driving the directional import of basic mitochondrial targeting signals across the TOM complex; the importance of these regions for driving import was termed the acid chain hypothesis [16]. The IMS region of hTom22 has features distinct from its fungal counterparts; notably, hTom22 lacks the acidic patch on the C-terminal IMS region that has been implicated in the acid chain hypothesis in fungal TOM complex translocation and contains a glutamine-rich domain that is also present in hTom20 [17]. Important to note is that the structure of the hTOM complex shows missing densities for residues 1–61 of hTom22 as well as residues 119–142, resulting in structural information only for the central region of full-length hTom22. Together, these data suggest a need for more specific studies to address the aspects of mitochondrial import that are distinct from the well-studied fungal counterparts. The initial discovery of hTom22 reported interactions with hTom20; however, the reported structure was unable to resolve this interaction despite seeing evidence for an interaction using cross-linked mass spectrometry (CL-MS) [18,19].

Tom22 has a broad interactome that extends beyond its TOM complex partners (Figure 5B). These interactions include proteins with roles in apoptosis, complex biogenesis, mitophagy, and mitochondrial protein translocation-associated degradation. Our reliance on the core TOM complex for structural information on Tom22 leaves open many questions on the structure of free Tom22 or the necessary conformational changes of Tom22 for interacting with other proteins. Additional structural information on full-length hTom22, particularly the N-terminal region, may provide structural insight on how Tom22 is able to recognize a wide variety of preproteins in the cytosol and the conformational rearrangements that may be necessary for recognition as well as the passing of preproteins to Tom40.

Structural support for Tom40 is provided by Tom22, but also bound to the Tom40 β-barrel are three β-barrel-associated subunits, Tom5, Tom6, and Tom7. Since the identification of Tom5 and Tom6 in humans, studies have revealed the roles of these proteins in creating stable intermediates for the integration of newly synthesized TOM complex components and how they work together to maintain various organizations of the TOM complex [19]. The TOM core complex is formed by Tom40, the β-barrel associated subunits, Tom5, Tom6, and Tom7, and Tom22. The human cryo-EM structure of the TOM complex reveals that Tom5, Tom6, and Tom7 interact directly with the Tom40 β-barrel (Figure 6A). Tom6 and Tom7 occupy opposite regions of the β-barrel, and Tom5 is situated at the opposite regions of the Tom22 bound segment of the β-barrel in the dimer structure. The human cryo-EM TOM complex shows that these components contain some interesting structural features: the presence of a kink in the helices of Tom6, Tom7 and Tom22; bending of the helical segments of Tom7 and Tom22 in a clockwise fashion around the β-barrel to orient them in a close enough position to suggest an interaction; Tom5 tilts in a fashion that allows it to run parallel to the tilted β-barrel, and finally, the Tom6 helical domain closest to the cytosol runs parallel to the OMM (Figure 6A,B).

Knockdown experiments that monitor the impact of these small subunits on the arrangements of the TOM complex demonstrate that Tom6 is vital for the formation of the tetramer TOM complex because knockdown leads to more dimers, and knockdown of Tom7 has the opposite effect and leads to more tetramers. In addition to the observed changes in the arrangement of the TOM complex, knockdown of hTom7 also results in increased import of hTom40, which further demonstrates the highly coordinated roles of each component of the TOM complex [20]. The antagonistic role of β-barrel associated subunits Tom6 and Tom7 in the biogenesis of the TOM complex has been observed in yeast. In yeast, Tom6 lends itself more to the assembly of the TOM complex while Tom7 is more critical for the disassembly, a prerequisite for incorporating new components into the complex. This is the opposite of what has been observed in humans where Tom6 is shown to promote the assembly of higher-order oligomers of the TOM complex and Tom7 promotes the disassembly [21,22].

The most successful expression strategy for TOM complex structural studies has been biased to identifying the core complex components previously described. Structures of the full TOM complex including transiently associated receptor proteins Tom70 and Tom20 would provide a structural understanding of where and how these receptor proteins associate to the core TOM complex and the structural rearrangements necessary to accommodate the interactions. Recombinant expression of components of the TOM complex in *E. coli* has resulted in high-resolution structural information only for the cytosolic domains of Tom70 [23] and Tom20 [24] (Figure 7A). Pull-down experiments using recombinant expression of the cytosolic domains of Tom70, Tom20, and Tom22 suggest that the cytosolic domains are able to interact independently with presequence peptides in a preferential manner, and while they demonstrate specificity for some substrates, there is a large degree of substrate overlap between the two loosely associated receptor proteins Tom20 and Tom70 [25]. Studies on ribosomes proximal to the mitochondria that are actively translating mRNAs that encode mitochondrial proteins show that knockdown of Tom70 results in an increase of ribosomes translating mRNAs of proteins designed to be recognized by Tom20 and a decrease of ribosomes translating mRNAs of proteins destined for Tom70, which suggests that there are factors that result in import and translation adaptation based on the mitochondrial need [26].

The crystal structure of rat Tom20 provides some insight into how Tom20 recognizes presequences [24]. The cytosolic domain of rat Tom20 was fused to the presequence of aldehyde dehydrogenase (ALDH) and crystal structures of two different linkers revealed that only two of the three necessary leucine residues in the observed consensus motif have strong contacts with Tom20. Mutagenesis studies of Tom20 showed two hydrophobic sites necessary for presequence recognition. The structural significance of the third leucine in the consensus motif still requires additional structural investigation. In addition, how Tom20 undergoes transitions between the observed conformations and the transmembrane contributions, if any, to this rearrangement is currently unknown. In addition to these questions, how Tom20 accommodates longer presequences and presequences that diverge from the presequence consensus motif requires further structural characterization.

Tom70 contains a sizable C-terminal domain located in the cytosol and a single transmembrane-spanning domain at the N-terminus. The cytosolic domain of Tom70 contains an internal targeting sequence binding pocket in addition to an Hsp70 binding pocket when the protein is in the dimeric form [27]. The biological relevance of the dimeric Tom70 has been questioned, mainly because the orientation of the cavities in the dimeric form occludes binding of multiple internal targeting sequences as well as of chaperone docking [28]. Studies of yeast Tom70 paralog, Tom71, suggest that in yeast, Tom70/Tom71 is able to exist in both an open and closed state and interconversion between these two states may allow for dimerization [28,29]. Studies of human Tom70 (hTom70) determined that the more active state of hTom70 is the monomeric form, even though dimers and other higher-order oligomers can be detected under particular conditions. Sequence divergence in the putative dimerization interface between yeast Tom70 and hTom70 may account for some of the observed differences in oligomerization and multimeric function between the two species [30]. Differences between the observed fungal Tom70 and Tom71 N-terminal domains may suggest that a similar conformational change is possible in the human Tom70.

De novo partial loss-of-function mutants of human Tomm70 genes have been isolated from two minor patients that presented with similar but not identical clinical symptoms of neurological impairment including movement disorders and coordination issues, decreased muscle mass and white matter abnormalities as evidenced by brain MRIs. The variants identified in both patients, p.Thr607Ile and p.Ile554Phe, are in the sizeable cytosolic C-terminal domain of Tom70. Rescue experiments using the variants in *Drosophila melanogaster* show that the mutants are not able to rescue lethality in *Tom70* null flies [31]. Tomm70 loss-of-function mutations were also identified in another patient that presented with developmental delay, anemia and lactic acid build-up. The mutations were identified to be p.Ala582Val and p.Thr265Met and were linked to lower levels of dimeric Tom70 in human cells, no identified monomeric Tom70, lower levels and lower activity of assembled Complex IV and decreased expression levels of Tom70, Tom40 and Tom22 [32]. The lack of observed monomeric Tom70 in both the reference cells as well as mutant cells indicates that more studies are necessary for our understanding of the relevance of the oligomeric state to the active form of Tom70. Together these data illustrate the need for further structure–function studies of full-length Tom70 as well as identified mutants so we gain an understanding of the molecular mechanisms by which these Tom70 mutants result in clinical symptoms.

The availability of structures only of the cytosolic domains also limits our understanding of the contributions of the transmembrane domain to oligomerization and protein–protein interactions.

Together, the human cryo-electron microscopy structure of the TOM complex in combination with structural information of the cytosolic domains of Tom70 and Tom20 provide an excellent starting point for our understanding of how the structural features of each subunit contribute to protein import. Further structural characterization is necessary to capture import intermediates of the human TOM complex to reveal distinct pathways, as seen in the fungal counterparts, as well as full length structures of Tom22, Tom20 and Tom70 to reveal how these receptors recognize a broad host of mitochondrial proteins.

### 2.2. Observed Structural Arrangements of the TOM Complex and Interplay with Other Mitochondrial Membrane Proteins

Advances in cryo-electron microscopy have provided structural insights on the TOM complex from *H. sapiens*, *N. crassa*, and *S. cerevisiae*. While this review focuses primarily on the complex in humans, the orthologs offer insight into the possible variations on organization of the TOM complex in humans. The various organizations of the TOM complex all have distinct proposed functions critical to the biogenesis of the TOM complex, as well as to adapting to various mitochondrial requirements. Structural support for these various arrangements of the TOM complex may be helpful in determining the relationship between various complexes and their roles in mitochondrial function and homeostasis, as well as to offer mechanistic information on how the inability to form a particular arrangement is a result of cellular stress and dysfunction. The diverse roles that components of the TOM complex play in mitochondrial biology and cellular homeostasis make them all-important targets for our understanding of human health and disease.

Due to each component of the TOM complex having such a wide variety of interaction partners, there must be some disassembly of mature TOM complexes or reservoirs of reduced TOM complex components that are available to interact with essential interaction partners under distinct conditions in various tissue types. In yeast, the ability of TOM complex subunits to exist when unbound to the mature TOM complex has been shown with Tom22 which binds to Por1, complexome profiling which reveals large (400 kDa+) complexes formed between components of the TOM complex and Por1, as well as competition experiments illustrating that precursors substrates utilize different arrangements of the TOM complex [33,34,35]. Together these data support the dynamic nature of the TOM complex subunits in yeast and reveal an important area of future study for understanding human TOM complex dynamics. Based on high resolution cryo-EM structural studies of the TOM core complex, two arrangements have been reported: A dimer as observed in the structure of *S. cerevisiae*, *N. crassa*, and *H. sapiens* that contains a dimer of Tom40, Tom5, Tom6, Tom7, and Tom22 (Figure 3A), and *S. cerevisiae* in a tetramer containing all five components which resemble a dimer of the previously described dimers having two copies each of Tom6, Tom22 and Tom5 at the dimer-to-dimer interface (Figure 3B) [36,37,38,39,40]. This tetramer was also observed in structural studies on the human TOM core complex however the resolution was much lower at 8.5 Å. How this tetramer functions and the conditions under which it forms require further investigation. Biochemical and low-resolution structures report two additional arrangements of the TOM complex in *N. crassa* and *S. cerevisiae*, a trimer containing the three copies of Tom40, Tom5, Tom6, Tom7, Tom22) and a dimer of Tom40 lacking Tom22, with two complete sets of the β-barrel associated subunits (Figure 3C) [40,41,42,43]. Overlap of the three reported structures of the dimeric TOM complex in *S. cerevisiae*, *N. crassa*, and *H. sapiens* demonstrate some structural differences. Superimposition of the three reported atomic resolution structures of the dimeric TOM core complex in *S. cerevisiae* and *H. sapiens* demonstrate a great degree of structural similarities with the *H. sapiens* structure offering the greatest degree of completion. The human structure offers a more complete model of Tom40, showing internal loops absent in the models of the fungal TOM complex. All three structures, however, were unable to resolve the IMS domain of Tom22 which may suggest a highly flexible domain.

The structures of the *S. cerevisiae* TOM core complex were reported at 3.1 Å and 3.8 Å by two independent groups and the human TOM core complex was reported at 3.4 Å. Similarities between all three dimer structures show bound lipid/detergent molecules between Tom40 β-barrels and flanked by Tom22, as well as two-fold symmetry and tilting of Tom40 β-barrels resulting in close proximity of the cytosolic regions of the β-barrels. Sequence differences between fungal and human TOM complex point to possible differences in preprotein import pathways such as the absence of the fungal C-terminal Tom40 segment that extends into the IMS, an extended IMS segment of human Tom7 that contains a negatively charged patch, and changes in the kink location of Tom22 resulting in human Tom22 being in closer contact with the Tom40 β-barrels and having a greater distance between the two helices near the IMS (Figure 3D).

Despite significant advances in our structural knowledge of the TOM complex, many questions remain such as: what is the physiological relevance of the observed states of the TOM complex, what factors determine interchange between various oligomeric states, what are the full-length structures of Tom70 and Tom20, how do Tom70 and Tom20 fit into the arrangement of the TOM core complex, and what oligomeric states are present in humans?

## 3. The Role of the TOM Complex Subunits in Human Health and Disease

### 3.1. The Role of Tom40 in Mitochondrial Homeostasis, Human Disease, and Therapeutics

The TOM complex is responsible for importing over 1000 members of the mitochondrial proteome in yeast and 1500 in humans. The TOM complex’s central pore is formed from Tom40, a 19 stranded β-barrel that contains mostly negatively charged regions except for a positively charged region near the IMS (Figure 4A). These electrostatic features are critical to recognizing and transporting presequences that are typically positively charged amphipathic α-helices [44,45]. In addition to the protein’s electrostatic features, the size of the β-barrel suggests that it is possible to import proteins in partially folded states: studies in yeast confirm that a particular preprotein, ADP/ATP carrier protein (AAC), translocates through the β-barrel in this manner [46].

In humans, two isoforms of Tom40 have been identified, Tom40A and Tom40B [44]. The most studied isoform, Tom40A, differs from Tom40B by containing an extended N-terminal region. In the rat homolog of Tom40B, it has been shown that the C-terminal helix is essential for targeting in a way that was not seen for the predominant isoform, Tom40A. How Tom40A and Tom40B interact to form complexes, import proteins, and the physiological functions of the two isoforms have not been well characterized in humans and this is an exciting area for further study. For the purposes of this review, we will focus on the better characterized Tom40A, referred to as hTom40 in humans.

While the bulk of studies on Tom40 have focused on its role in yeast, mitochondrial import studies of hTOM40 reveal that in addition to its primary role in preprotein import, Tom40 also serves as a passage for viral proteins into mitochondria and can do so without the assistance of other TOM complex receptor proteins [47]. An example of viral import comes from the Influenza virus protein PB1-F2. PB1-F2 variants containing a C-terminal polypeptide are imported into mitochondria and localize to the IMS. The mitochondrial-targeted variant of PB1-F2 in the IMS results in a reduction of the mitochondrial membrane potential, morphological changes that result in fragmented mitochondria, and inhibition of the RIG-I signaling pathway. The RIG-I signaling pathway is a major pathway necessary for the recognition of RNA viruses and activating the immune response [48]. Influenza A viral variants that encode a truncated version of PB1-F2 that does not target mitochondria are associated with lower pathogenicity, indicating the importance of mitochondrial targeting to increased pathogenicity [47]. Inhibition of mitochondrial targeting of PB1-F2 may serve as a potential therapeutic strategy for Influenza A. In addition to RNA viral proteins being targeted to mitochondria, evidence for DNA virus Acanthamoeba polyphaga mimivirus (Mimivirus) protein VMC1 shows a similar mitochondrial import through Tom40 with the aid of Tom70 [49]. Together these data suggest that mitochondrial targeting of viral proteins may be significant through multiple pathways that aid in viral infection. Tom40 as a viral target makes its structure and function important for our understanding of viral pathogenesis and as a target for therapeutic development.

Tom40 also plays several roles in autophagy and mitochondrial homeostasis through interactions in mitochondria-endoplasmic reticulum contact sites (MERCS) [50]. The critical role of MERCS in essential processes implicates them in cancer, neurodegenerative diseases, innate immunity, and aging. MERCS are regions where there the mitochondrial outer membrane is found near ER membranes without any fusion of the membranes. MERCS contain regions of specialized proteins and lipids essential for cellular functions such as lipid transfer, autophagy, and calcium homeostasis. MERCS have a number of tethering complexes that include various OMM proteins such as VDAC, MFN2, and Tom40. Notably, Tom40, in concert with Tom70, plays a vital role in MERCS through its interactions with Atg2, an autophagosome-related protein responsible for lipid transfer, and BAP31, an integral ER membrane protein implicated in apoptosis and mitochondrial homeostasis [50,51,52,53]. Atg2 orthologs, Atg2A and Atg2B, are both critical to the maturation of premature autophagosomes and important to this process is their localization to MERCS. The importance of autophagy to the inhibition of tumor formation implicates Atg2 in cancer studies. A C-terminal region of Atg2A that is sufficient for localization of Atg2A to MERCS, termed the MERCS localization domain (MLD), is the region that interacts with hTom40 and with the aid of hTom70 is responsible for Atg2 targeting to MERCS for phagophore expansion [52]. BAP31, also involved in MERCS, interacts with hTom40 to aid in the localization of NDUFS4, a nuclear-encoded mitochondrial protein that is a subunit of Complex I in the IMM. BAP31-Tom40 complex interacts with both NDUFS4 precursors to aid in translocation from the cytosol into mitochondria and mature NDUFS4 to the IMM by triggering mitochondrial oxygen consumption. This interaction was shown to be decreased under ER stress conditions, stimulated by addition of ER stress inducer brefeldin A, indicating that the role of Tom40 and other interaction partners varies in different cellular states [53]. In addition to the reliance of NDUFS4 mitochondrial localization on the BAP31-Tom40 interaction, NDUFB11, an accessory subunit of Complex I, was also reduced in BAP31 depletion cells. Together these data suggest the importance of this interaction in the assembly of Complex I, the role of BAP31-Tom40 in mitochondrial import and could be important for our understanding of diseases related to Complex I dysfunction.

Tom40 carries out its role in mitochondrial homeostasis through interactions with other proteins such as Atad3a, a mitochondrial AAA-ATPase. Atad3a can cause neurodegeneration through oligomerization, serves as a stem cell regulator, and through facilitation of Pink1 import is critical to mitophagy. Atad3a interacts with Tom40 and translocase of the inner membrane [54] complex component Tim23 to promote the import of Pink1, which prevents recruitment of Parkin that can trigger mitophagy [55].

Tom40 is responsible for the mitochondrial import of α-Synuclein, a presynaptic protein, which is a significant component of Lewy Bodies and Lewy neurites which are hallmarks of Parkinson’s Disease (PD). Decreases of Tom40 levels in PD patients have been observed, which were not seen for other components of the TOM complex. These decreases are seen with wild type α-Synuclein and A53T α-Synuclein, one of two mutants known to cause early-onset PD. The cause for the observed lack of decrease in Tom40 in the other mutant A30P requires further study. Overexpression of Tom40 in α-Synuclein accumulating cells serves to rescue mitochondrial quality by decreasing some α-Synuclein accumulation, reactive oxidative species (ROS) production, and mitochondrial DNA damage [56]. This may provide information on how modulating Tom40 levels may serve as a starting point for a therapeutic strategy for PD.

Tom40 has also been implicated in other neurodegenerative diseases such as Late-onset Alzheimer’s disease (LOAD) and Huntington’s disease (HD). Further study of the role of Tom40 in these disease states will advance our understanding of the interplay between mitochondrial protein import and function. The role of Tom40 in cancer, though less studied than its function in neurodegenerative diseases, also reveals that in epithelial ovarian cancer, higher levels of the protein correlate with poorer survival outcomes [37]. Further study on the role of Tom40 in cancer is necessary to increase our understanding of tumor types that do rely on oxidative phosphorylation in mitochondria in place of the Warburg effect often observed in tumor cells.

The abundance and prevalence of Tom40 across human tissue types make it unsurprising that there are currently no therapeutics that target Tom40 directly; however, our understanding of its function in the context of stress and disease may inform how we may develop targets for Tom40 interaction partners [57].

### 3.2. The Role of Tom22 in Mitochondrial Homeostasis, Human Disease, and Therapeutics

The structural reliance of the core TOM complex on hTom22 makes it no surprise that Tom22 deficient cells demonstrate impaired viability in mammalian cells and yeast [58]. The essential role of Tom22 in animal cells has been demonstrated in various cell types. Here we will discuss its role in adrenal tissues, gonadal tissues, and hepatic tissues. Mutation studies in zebrafish reveal that *tom22* mutants have impaired hepatocyte function resulting from apoptosis that leads to irregular hepatic tissue size and organization. Interestingly, this mutation showed no obvious morphological defects in cardiac tissue despite its heavy reliance on proper mitochondrial function; the mutant Tom22 was adequately targeted to mitochondria, suggesting that the basis for these morphological differences is not due to improper targeting and occurs further downstream [59]. In adrenal and gonadal tissues, which require steroid synthesis, studies show that knockdown of Tom22 results in inhibition of 3 β-hydroxysteroid dehydrogenase 2 (3 βHSD2) expression. The interaction between Tom22 and 3 βHSD2 results in the formation of a more extensive metabolic protein complex in concert with other proteins, which is responsible for the catalysis of pregnenolone to progesterone. Together these data reflect the essential role that Tom22 plays in addition to protein import [60].

Approximately 1100 proteins of the ~1500 proteins imported by the TOM complex rely on hTom22 for mitochondrial import. Of the 1100 proteins for which hTom22 serves as a receptor, Bax is the one that is most relevant for hTom22’s role in apoptosis [61]. Bax is a member of the Bcl-2 family, which includes a host of both pro- and anti-apoptotic proteins. Bax is essential for apoptosis which has made it highly targeted in cancer therapeutics [62]. Studies on the mechanism of Bax insertion reveal that once Bax is integrated, it can also serve as a receptor for additional Bax molecules, which would speed the formation of Bax oligomers and supports the all or nothing mechanism of apoptosis [63]. Of interest is the absence of Tom22, Tom20 and Sam50 in high-density Bax rings formed during apoptosis. How these proteins become excluded from these Bax rings may provide insight into the molecular mechanisms of pore formation during apoptosis [64]. Inhibition of Tom22–Bax interactions through antibody blocking, or Tom22 knockdown, results in inhibition of Bax-mediated apoptosis. Studies of this interaction in yeast also reveal that human Bax interacts with other components of the TOM complex: Tom70 and Tom40 [65]. The participation of Tom22 in the apoptotic process is also seen in HIV in vivo models. HIV-1 protease (PR), which localizes to mitochondria, triggers apoptosis, and one possible mechanism is through the cleavage of Tom22, VDAC, and adenine nucleotide translocator (ANT) [66]. Computational studies on the structural roles of mitochondrial import proteins in apoptotic signaling suggest the possible formation of a complex between Tom40–Tom22–Bax and tBid, a membrane-targeted ligand involved with apoptosis; however, further structural studies capturing these interactions are necessary to consider ways these potential interactions could be therapeutically manipulated [67].

One small molecule that has been shown to target Tom22 is celastrol, a proteasome inhibitor found initially in *Tripterygium wilfordii*, that has been studied for its role in various inflammatory diseases, cancer, and neurodegenerative diseases. It has been demonstrated that celastrol inhibits gastric, ovarian, and colorectal cancer growth by forcing cell cycle arrest. During ER stress-mediated apoptosis, celastrol induces apoptosis through the previously described interaction between Bax and Tom22, and also results in increased expression of both proteins, consistent with previously described data [68]. The role of Tom22 in apoptosis requires further study to understand how its vast interactome may reveal opportunities for therapeutic intervention.

Decreased expression of Tom22 is also seen in epithelial cells under high glucose conditions, which results in apoptosis and other phenotypes specific to the cell type. In this study, the relationship of Tom22 to hyperglycemia was probed to determine the link between mitochondrial protein import components and the development of diabetes-related vascular complications. In human umbilical vein endothelial cells, Tom22 knockdowns show smaller and more fragmented mitochondria, which is a phenotype rescued by the overexpression of Tom22. Mfn1, located downstream of this glucose response pathway, is also downregulated during high glucose and serves as an interaction partner for Tom22 [69].

Despite the vast and complex interactome of Tom22, studies in specific tissue types reveal the specialized roles Tom22 can play in cells. For example, cardiac cells, which have a heavy reliance on ATP produced through oxidative phosphorylation (OXPHOS), are an excellent case study on how tissue-specific differential expression of genes can rely on Tom22 for import. The heavy economic burden of cardiovascular disease paired with the fact that cardiac cells can produce upwards of 6 kg of ATP per day, has resulted in many studies on the molecular underpinnings of mitochondrial function and mitochondrial protein import in cardiac tissue. Mitochondrial BK_Ca_ channels (mito BK_Ca_), which are calcium-activated potassium channels, are expressed in heart and brain tissue and carry out a cytoprotective function in both tissues [70]. Mito BK_Ca_ interacts with several TOM complex components, notably Tom22, Tom40, and Tom70, and is imported through its interactions with these components [71]. Other non-import-related roles of the interaction are possible and require further investigation. Mito BK_Ca_ may follow one of several import routes that rely on Tom22 for protein import. Broadly speaking, preproteins can be recognized first by Tom20 and then passed to Tom22 for the subsequent steps of import. This pathway has been well characterized in yeast and is biased to proteins that contain an N-terminal mitochondrial targeting sequence. Preproteins containing internal mitochondrial targeting sequences have been shown to primarily interact with Tom70 before being transferred to Tom22 for the remaining steps of import. In fungal studies, there can be some overlap of substrates between Tom70 and Tom20 before being handed off to Tom22 [72]. This demonstrates that the path preproteins take before arriving at Tom22 can reveal some information on its targeting sequence. In addition to the described import roles of Tom22 in cardiac tissue, it is possible that Tom22 forms complexes with other mitochondrial proteins to carry out distinct regulatory functions as seen in yeast with its interactions with the VDAC ortholog Por1. Additionally, in yeast, it has been demonstrated that Tom22 is able to interact with components of the IMM complex TIM23 resulting in a stable TIM-TIM23 super complex [73]. Contact sites between the TOM complex and the TIM complex have been shown to be integral for import efficiency, and crosstalk between the two complexes is increased during active translocation however the interaction is present when no precursors are present suggesting a possible conformationally change resulting in increased contacts to TIM23 [74]. Differences between the mammalian and fungal TIM23 complex include absent domains on TIM23 complex subunit Tim50 which has been implicated in forming contacts with the TOM complex in yeast [75]. Further investigation is necessary to elucidate the mechanistic difference in protein translocation between mammalian and fungal species as it pertains to the interactions between the TOM complex and TIM23 complex.

Using a biochemical approach, researchers recently described the formation of a three-protein complex between Tom22, aldosterone synthase (P450C11AS), and novel protein steroidogenic acute regulatory protein (StAR) that demonstrates the role of this triprotein complex in aldosterone synthesis. Interestingly, its localization has been found at contact sites between the OMM and the IMM, suggesting that Tom22 can interact with additional proteins in the IMM [76]. Further investigation of the additional import independent roles of Tom22 in cardiac cells is required.

Tom22 also plays major roles in mitochondrial quality control. The nexus of the high energetic demands of neural tissue with the targeting of aggregation-prone proteins to mitochondria creates an opportunity for a deeper molecular understanding with the goal of therapeutic development to address multiple neurodegenerative diseases [76].

Tom22 is involved in mitophagy-related protein complexes that include Pink1 [77]. To describe the importance of Tom22 to this process, we must first discuss broadly how mitochondrial quality control is regulated by Pink1 and Parkin, the associated protein mutations involved in familial PD, and proposed mechanisms for sporadic PD. Familial PD has been linked to mutations in Parkin, the gene associated with Park2, an E3 ubiquitin ligase, and PINK1, a kinase. Mutations in these proteins lead to dysregulation of the Pink1/Parkin mitophagy pathway at various steps [78,79]. Recognition of healthy mitochondria relies on the maintenance of the membrane potential at the IMM, which is necessary for import. In healthy mitochondria, Pink1 is imported into the matrix, where it undergoes cleavage by mitochondrial proteases and undergoes complete degradation by the proteasome. Dissipation of the membrane potential indicates loss of mitochondrial function and leads to accumulation of Pink1 on the OMM. This triggers the recruitment of Parkin as well as activating the kinase activity of Pink1, which results in signal amplification, polyubiquitylation of OMM proteins, and results in varying degrees of degradation depending on the extent of damage. The accumulation of Pink1 results in interactions with various components of the TOM complex, which includes Tom22 [77,80]. Tom22 is one of the proteins that undergoes ubiquitylation during mitochondrial stress triggered by chemicals that lead to membrane depolarization [81].

Studies in yeast also demonstrate that Tom22 can be degraded by alternate mechanisms requiring dislocation from the membrane to the IMS and the formation of a complex involving AAA ATPase Yme1 [82]. This mechanism is essential for monitoring the quality of components of the TOM complex on the IMS side, and other mechanisms that rely on other components of the TOM complex are responsible for monitoring translocation fidelity at the cytoplasmic side [83]. Though recent discoveries have provided insight on mitochondrial protein quality control mechanisms in yeast, more research is necessary to understand these processes in humans and how they interplay with other broader mitochondrial quality control mechanisms.

Studies on other post-translational modifications of Tom22 show how these can play a role in quality control in mammalian cells. In addition to degradation-related ubiquitylation of Tom22 in mammalian skeletal muscle, Tom22 undergoes phosphorylation by CSNK2. In yeast, this phosphorylation by CSNK2 yeast ortholog CK2 has been shown as critical for the biogenesis of the TOM complex [84]. However, in mammalian cells, the phosphorylation of Tom22 has no import or biogenesis-related function and instead is involved with Pink1/Parkin-related mitophagy by increasing import of Pink1. The loss of this phosphorylation impairs Pink1 import and promotes mitophagy [85]. Together these data are reflective of how Tom22’s broad range of interaction proteins allows it to carry out distinct functions in many tissue types and under various stress conditions.

### 3.3. The Roles Tom5, Tom6, and Tom7 in Mitochondrial Homeostasis, Human Disease, and Therapeutics

As with much of our understanding of the TOM complex, the bulk of our understanding of the roles of Tom5, Tom6, and Tom7 stems from studies in yeast. Studies in yeast reveal that Tom5 contains a critical proline residue similar to what is observed in Tom7 that is important for the targeting and integration of Tom5 into the TOM complex [86]. In yeast, Tom5 is involved in the acid chain hypothesis and serves as an essential receptor for the import of proteins. Despite differences between hTom5 and fungal Tom5, studies in yeast have offered insight into the role of Tom5 in human disease and genetic disorders. Specifically, studies of Taz1, the yeast ortholog of human tafazzin, which is involved in Barth Syndrome, reveal that in yeast, Taz1 import is reliant on Tom5 [87]. Knockdown of Tom5 in yeast results in significant growth defects and is shown as essential for viability, which is different than has been observed with hTom5 [20]. The role of yeast Tom5 in import is mediated through interactions between the N-terminal helix segment of Tom40 and the proximal C-terminus of Tom5 near the IMS. This region is absent in hTom40, which suggests additional work is necessary to determine precisely how hTom5 functions in protein import [14]. Studies on the interaction partners of fungal Tom5 reveal that there is a pool of Tom5 that interacts directly with the SAM complex to aid in the assembly and folding of fungal Tom40 [88].

Though there are fewer studies on hTom5 interaction partners and the importance of these interactions in disease, hTom5 has been used as a fusion protein partner to target tumor suppressor p53 to mitochondria to study how mitochondrial localization of p53 impacts proliferation of human non-small lung cancer cells, a cell type that is not subject to p53 mediated apoptosis [89]. Further investigation is necessary to determine what cancer gene therapy possibilities there are through capitalizing on mitochondrial targeting to trigger apoptosis.

As discussed with hTom5, there is a similar dearth of studies on the role of hTom6 in disease and its interactome. Observations in yeast reveal that Tom6, in concert with SAM complex component Sam37, aids in the assembly and integration of Tom40 into the TOM complex [90]. Based on available data, yeast 2-hybrid screens reveal an interaction between Tom6 and divalent metal transporter 1 (DMT1) using a human kidney cDNA library. DMT1 plays critical roles in iron transport, and inhibition of transport activity has been shown to result in cancer stem cell death [91]. It is interesting to note that in this study, DMT1 did not show any interaction with any other components of the TOM complex, which necessitates further exploration to determine if this suggests an additional role aside from import for the interaction between hTom6 and DMT1 [92].

Data on hTom7 offer a picture of a distinct interactome from the other two β-barrel-associated subunits (Figure 6C). Studies on the assembly and integration of Tom7 suggest that unlike fungal TOM complex assembly, Tom7 is first integrated through a hTom22-free intermediate complex; however, assembly intermediates of hTom7 into mature TOM complexes do seem to demonstrate variability between cell types, which may be indicative of complex rearrangements according to the metabolic needs of the cell [93]. Studies on the localization of hTom7 show that fusion of GFP to the IMS facing C-terminus completely abrogates proper targeting of hTom7, which could be explained by steric hindrance or interference of the mitochondrial targeting sequence. hTom7 is also implicated in Pink1/Parkin mediated mitophagy as critical for the recruitment of Parkin, its OMM accumulation, and kinase activity. In the hTom7 deficient cells, PINK1 accumulation does not occur even under depolarization conditions. This phenotype is rescued by knockdown of IMM protease OMA1, which may serve as a druggable target for the treatment of PD. [94]. Together these data demonstrate that the β-barrel associated subunits of the human TOM complex are an area of mitochondrial import that requires further exploration to determine the nuances of how the human TOM β-barrel associated subunits function differently from their fungal counterparts.

### 3.4. The Roles Tom20 and Tom70 in Mitochondrial Homeostasis, Human Disease, and Therapeutics

Tom20 (often used as a mitochondrial marker) and Tom70 serve as receptors for the initial binding of the bulk of the mitochondrial proteome and are also only transiently associated with the core TOM complex, so it should come as no surprise that these membrane proteins serve diverse roles in mitochondrial biology and are critical to our molecular understanding of a host of diseases in a variety of human cell types. Here we discuss the roles these proteins play in mitochondrial homeostasis, our structural understanding of these proteins, current disease models that implicate these two proteins, and therapeutics currently in the pipeline that target these components of the TOM complex.

Tom70 critical mitochondrial functions are not only limited to protein import. Tom70, in concert with Tom20, has been implicated in complexes that contain Pink1 following depolarization; however, this has not been consistent across experimental reports which have varying degrees of sensitivity for detecting transient interactions [77,95]. Studies have shown that the interaction between Tom20 and Pink1 can be disrupted by celastrol, which inhibits Pink1 mitophagy providing evidence for the Pink1–Tom20 interaction and its importance in Pink1/Parkin mediated mitophagy [96]. In HEK293T cells, it was demonstrated that mitochondrial import is monitored in order to trigger mitophagy in cases of import blockage. This is mediated by interactions between Tom70 and Parkin. Interesting to note is that several common Parkin mutants reduce the interaction between Tom70 and Parkin [97]. In addition to direct interactions with Parkin, Tom70 is critical for Pink1 import, which offers some mechanistic insight on Pink1 import [98]. Together, these data implicate Tom70 as critical for mitochondrial quality control. Further investigation is necessary to identify the exact physiological conditions under which Tom70 is included in depolarization-induced Pink1 complex formation [76,99]. Recently, studies of Pink1 kinase activity under non-depolarization conditions revealed that components of the TOM complex (Tom20, Tom22, and Tom70) undergo Parkin-dependent ubiquitylation and USP30 deubiquitylation [81,100]. This ubiquitylation of Tom70 results in increased import reliance on Tom20 dependent pathways. The effects of various types of ubiquitin chains on import efficiency or mitophagy require further investigation to determine the regulatory roles ubiquitylation may play in human mitochondrial import. In addition to ubiquitylation of Tom70 as a post-translational modification, we also know that hTom70 undergoes extensive phosphorylation, and much like ubiquitylation, we do not have a good grasp on the roles of these modifications play in Tom70 function and regulation [101]. It is evident, however, that various familial mutations of *PINK1* and *PARK2* do reduce import efficiency connecting mitochondrial import to the pathology of PD. Studies in yeast reveal that Ubx2, a protein involved in ER-associated degradation, is responsible for the recruitment of AAA ATPase Cdc48 for the removal of clogged preproteins from the TOM complex. Tom70 serves as the primary receptor for the import of Ubx2 [83,102]. More exploration is necessary to elucidate what, if any, are the mitochondrial protein translocation clearance mechanisms present in mammals.

In addition to Tom70’s role in mitophagy, it has been implicated in apoptosis through interactions with Tom20 and pro-apoptotic protein Bim in mouse embryonic fibroblasts and HeLa cells [103]. Bim contributes to the activation of Bax and promotes apoptosis. It is important to note that in Tom70 knockdowns, Bim was still localized to mitochondria and was able to promote apoptosis, even when all three receptors Tom70, Tom20, and Tom22, were knocked down, suggesting that Bim import is independent of TOM complex receptors. The function of the Bim–Tom70 and Bim–Tom20 interactions require further investigation. In addition to Tom70–Bim interactions, Tom70 has been implicated in apoptosis in the context of Hepatitis C Viral (HCV) infection [104]. Following HCV infection, Tom70 expression was increased, resulting in cells that were not sensitive to TNF-α-mediated apoptosis. Through exploring the cause for this increased expression, it was determined that Tom70 interacts with HCV non-structural protein 3 (NS3) protease. Knockdown of Tom70 results in increased apoptotic activity and decreases in both NS3 and Bcl2 related protein Mcl-1 [105]. This function of Tom70 in stabilizing both NS3 from HCV and Mcl-1, which is implicated in promoting tumor cell survival, may provide the link between hepatocyte tumorigenesis and chronic HCV infection. Tom70 also has a broader role in RNA viral infection recognition through interaction with the mitochondrially localized protein MAVS which, together with other cytosolic proteins, form the MAVS signal complex following infection [106,107]. The MAVS signaling complex is formed on the OMM following recognition of viral cytosolic RNA by RIG-1/MDA5 and then goes on to trigger kinase activation, which results in the production of IFN-b through the phosphorylation of interferon regulatory factor 3 (IRF3). Tom70 serves to increase the activation of IRF3 and its downstream effects when exogenously expressed and as expected, knockdown of Tom70 attenuates IRF3 activation. Tom70 carries out this function through the recruitment of important downstream signaling molecules to the mitochondria through interactions with Hsp90, which is essential for IRF3 activation. This interaction between Tom70-Hsp90-IRF3 also mediates Sendai virus-mediated apoptosis through interactions with pro-apoptotic protein Bax [106,107]. Tom70’s role in viral protein interactions has come into the spotlight recently following the extensive study of the coronavirus family of viruses due to the 2020 pandemic. These studies revealed SARS-CoV-2 viral protein Orf9b interactions with Tom70 (Figure 7A) [108]. Structural characterization of this interaction has provided the first structure of human Tom70 in a cryo-EM structure of Orf9b bound to the substrate-binding region of hTom70, where Orf9b must undergo a conformational change from the entirely β-sheet structures seen when unbound to an α-helical region when bound to hTom70 [109]. Tom70–Orf9b inhibits IFN-b activation [110]. Further exploration of the functional significance of this interaction is necessary and may serve as an interesting viral therapeutic strategy. A particular steroid that has been used in patients hospitalized with COVID-19, dexamethasone, has been shown to upregulate Tom70 and Tom20 in kidney cells and, in this context, reduces mitochondrial injury [111,112]. How dexamethasone modifies mitochondrial import in a viral infection context requires additional investigation.

In addition to its many mitochondrial roles, Tom70 is clustered at MERCS [113]. In these specialized regions, Tom70 interacts with inositol triphosphate receptor 3 (IP3R3). IP3R3 clusters at MERCS and is recruited to mitochondria by Tom70 and contributes to Ca^2+^ propagation from ER to mitochondria [114]. This interaction with IP3R3 and the reliance of the Krebs cycle on Ca^2+^ directly links Tom70 to mitochondrial respiration in a manner unrelated to its role as a protein import receptor. Further investigation of its role in mitochondrial calcium transfer is necessary to determine the full range of its role in MERCS.

Tissue-specific mitochondrial studies reveal the unique roles of Tom70. Tom70 plays a critical role in cardiac pathology through its properties that impact oxidation and apoptosis. Cardiac hypertrophy is the enlargement or thickening of cardiac muscle. Tom70 expression is reduced in cardiac hypertrophy cell culture and zebrafish models, as well as in patient hypertrophic cardiac samples [115]. Reduction of Tom70 results in several molecular consequences, such as a reduction of optic atrophy-1 protein (Opa1), which is heavily involved in mitochondrial morphology, and the decrease of Opa1 results in an increase of oxidative stress. Opa1 has also recently emerged as mitochondrial target for cancer therapeutics, such as MYLS22, due to its essential roles in mitochondrial dynamics and energy metabolism [116]. Tom70 and Opa1 show a direct interaction in pull-down assays. The import of Opa1 is also reduced in Tom70 deficient cells. Tom70 expression shows a protective effect against cardiac hypertrophy. This is also seen in cardiac myopathy, and the phenotype can also be rescued by restoring Tom70 levels [117]. Tom70 is additionally involved in mitigating myocardial infarction and ischemia/reperfusion injury through interactions with melatonin and mitochondrial calcium uptake 1 (MICU1) protein [118,119]. In both contexts, Tom70 deficiency resulted in increased injury, which could be restored with Tom70 supplementation. Tom20 levels are also reduced under ischemic conditions and can be rescued by ischemic preconditioning; however, less is known about the relationship between ischemia and Tom20 expression [120]. Together, these data provide evidence for Tom70 expression restoration as a viable therapeutic strategy for a host of cardiac diseases that result in cardiac failure; however, more information is necessary on the mechanism of Tom70’s effect on ROS levels.

While there is a great degree of overlap between the Tom70 interactome and the Tom20 interactome, each receptor protein has unique interacting partners involved in disease pathogenesis and roles outside of their primary function of import (Figure 7B). The most notable Tom20 specific interacting partner is α-Synuclein. High-affinity binding of α-Synuclein to Tom20 prevents its interaction with Tom22 resulting in defects in protein import in PD [121]. Excitingly, overexpression of hTom22 is able to rescue the observed phenotype. The Tom20 role in import has also been linked to proteins critical to infection and pathogenicity. Tom20 imports the mitochondrially targeted p34 subunit VacA, a toxin produced by *Helicobacter pylori.* Upon Tom20 dependent import into mitochondria, p34 forms a large pore that serves as an anion channel [38]. Tom20 also interacts with aryl hydrocarbon receptor-interacting protein (AIP) which together are responsible for the import of survivin, a protein involved in apoptosis inhibition and cell division, into mitochondria. Knockdown of Tom20 in HeLa cells abrogated survivin accumulation and allowed these cells to be sensitive to apoptosis by staurosporine [39]. An in-depth understanding of Tom70’s functions independent of Tom20 will provide greater insight into the nuances between these two receptor proteins and the same can be said about the reciprocal Tom20-independent functions of Tom70.

## 4. Conclusions

While advances in our understanding of the yeast TOM complex have served as the foundation for much of the knowledge of mitochondrial protein import in humans, the increased complexity of humans reveals some distinct differences in how components of the TOM complex are able to function (Figure 2). Targeted studies on the TOM complex components in humans reveal a broad interactome that serves the various needs of mitochondria across cell types and disease states. Continued research on how mitochondrial import is regulated to serve cellular energetic demands in cell types that are heavily reliant on mitochondria will provide insight into how mitochondrial import is influenced by stress and mitochondrial dysregulation. An in-depth understanding of the roles TOM components play outside of import would provide molecular insights for the pathogenesis of many cancer types, neurodegeneration, viral immunity, and mitochondrial genetic disorders that would bolster mitochondria as a therapeutic target. This review primarily focuses on TOM complex interactions; however, the remaining protein import, sorting, and assembly machinery of mitochondria can also provide similar insights into how the mitochondrial proteome responds to disease.

## Figures and Tables

**Figure 1 cells-10-01164-f001:**
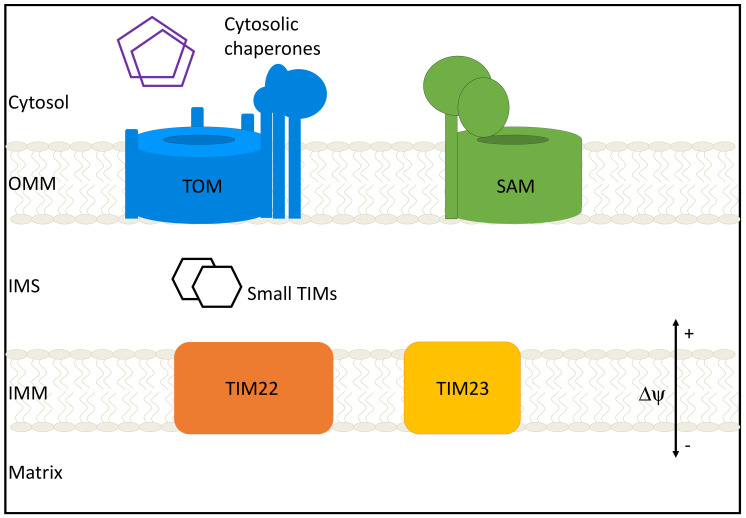
Mitochondrial protein import machinery. The bulk of protein import occurs through the translocase of the outer membrane (TOM) complex. Cytosolic chaperones aid in this process by preventing aggregation of preproteins. After traversing the TOM complex and entering the intermembrane space (IMS), proteins can follow one of several general import pathways: β-barrel proteins are inserted into the outer mitochondrial membrane (OMM) by the sorting and assembly machinery (SAM) complex; Carrier proteins are inserted into the IMM by TIM22; Proteins destined for the inner mitochondrial membrane (IMM) or matrix are imported and integrated by TIM23; small TIMs support protein transfer from IMS to complexes.

**Figure 2 cells-10-01164-f002:**
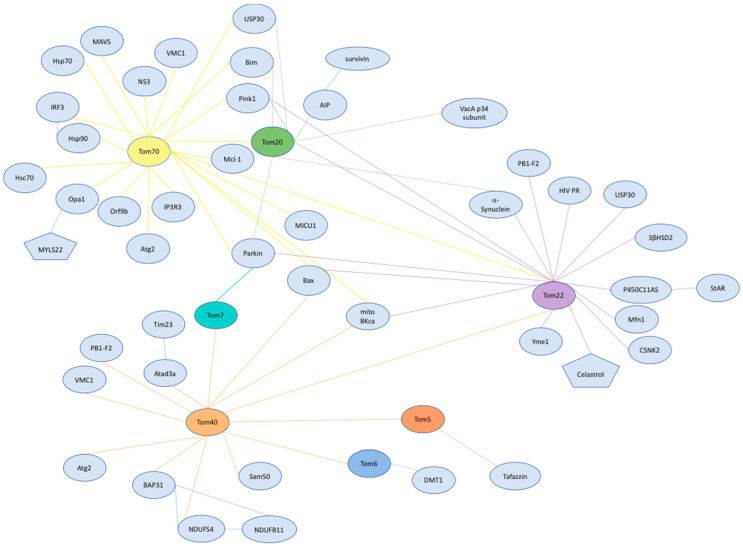
TOM complex interactions. Interaction map of TOM complex subunits demonstrating the vast network that has functions in import, immunity and infection, mitochondria-endoplasmic reticulum contact sites (MERCS), cancer, metabolism, apoptosis, and neurodegenerative diseases. Ovals represent proteins, and pentagons represent small molecules. TOM complex subunits are colored for ease of understanding. Tom40 (orange), Tom22 (purple), Tom5 (vermillion), and Tom6 (light blue), Tom20 (green) and Tom7 (seafoam green).

**Figure 3 cells-10-01164-f003:**
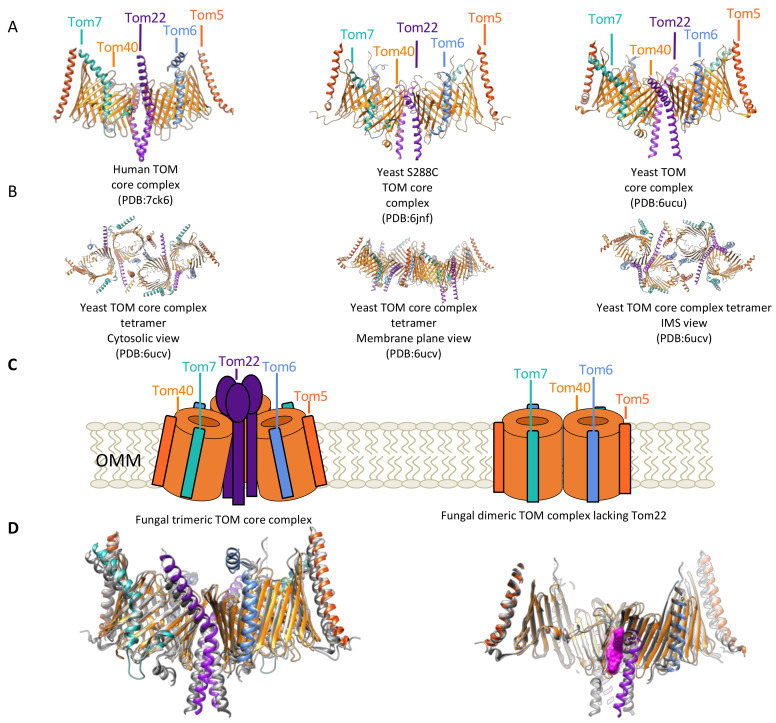
Arrangements of the TOM complex. (**A**) Structures of the human TOM core complex (**left**, PDB:7ck6) illustrating high degrees of structural homology with some variations within the subunits and the dimeric form of two separately reported *S. cerevisiae* TOM core complex structures (**middle**, PDB:6jnf; **right**, PDB:6ucu). All three structures show two Tom40 β-barrels (orange), with two Tom22 (purple) helices between the β-barrel, opposite the Tom22 dimers on both β-barrels is a curved Tom5 (vermillion), and on opposite sides of the β-barrel between Tom5 and Tom22 are Tom6 (light blue) and Tom7 (seafoam green). (**B**) Structures of tetrameric yeast TOM core complex. (**left**) Cytosolic view of yeast TOM core complex tetramer, illustrating Tom6 role in mediating the formation of the tetramer in yeast. (**middle**) Membrane plane view of yeast TOM core complex illustrating the tilt of the Tom40 β-barrels and how this results in the curvature of the tetrameric TOM complex. (**right**) IMS view of yeast TOM core complex showing proximity of IMS domains of β-barrel associated subunits to Tom40 domains that extend into the IMS. Notably, the N-terminal region of Tom40 and Tom5, and the C-terminal region of Tom40 and Tom7 are all in close proximity. (**C**) (**left**) Cartoon representation of fungal TOM core complex trimer, (**right**) fungal Tom40 complex dimer lacking Tom22 based on crosslinking and biochemical data. (**D**) (**left**) Superimposition of fungal TOM complexes (gray) and human TOM complex (colored), (**right**) Superimposition of fungal TOM complexes (gray) and human TOM complex with lipids/detergent in magenta (space-filling model). Superimposition highlights closeness of IMS region of hTom22, extended helical region of hTom6 that runs parallel to the OMM near the cytosol, and extended loop of Tom7 near the IMS.

**Figure 4 cells-10-01164-f004:**
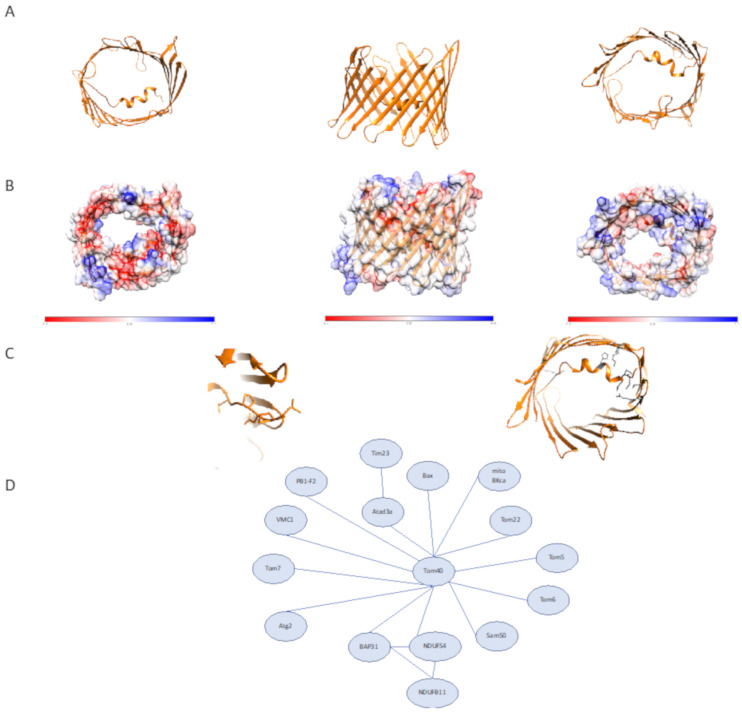
Human Tom40 structural features and interactome. (**A**) Structure of human Tom40 (PDB:7ck6). (**left**) Cytosolic view. (**middle**) Membrane plane view. (**right**) IMS view (**B**) Surface electrostatic features of human Tom40. (**left**) Cytosolic view. (**middle**) Membrane plane view. (**right**) IMS view (**C**) The N-terminal coil of human Tom40. (**left**) visible coil region preceding the α-helical segment of N-terminal Tom40 region. (**right**) Interactions between Tom40 lumen and N-terminal region, cytosolic view with relevant residues shown in gray. (**D**) Tom40 interactions as described. Distances do not represent interaction strength.

**Figure 5 cells-10-01164-f005:**
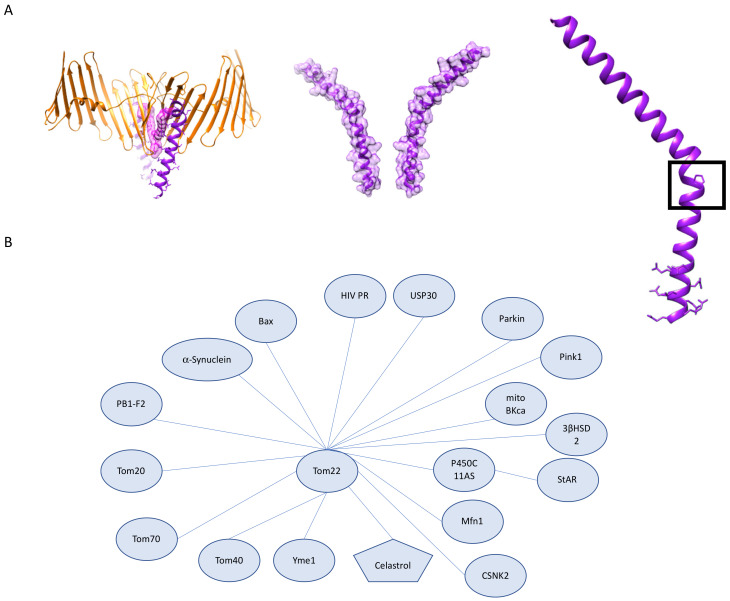
Tom22 structural features and interactome. (**A**) Structure of human Tom22 (PDB:7ck6). (**left**) Tom22 interactions with lipid (magenta, space-filling model) and Tom40 β-barrel exterior. (**middle**) Tom22 dimer (space-filling model). (**right**) Proline responsible for Tom22 kink shown in black box as well as the Q rich C terminal motif. (**B**) Tom22 interactions as described. Distances do not represent interaction strength.

**Figure 6 cells-10-01164-f006:**
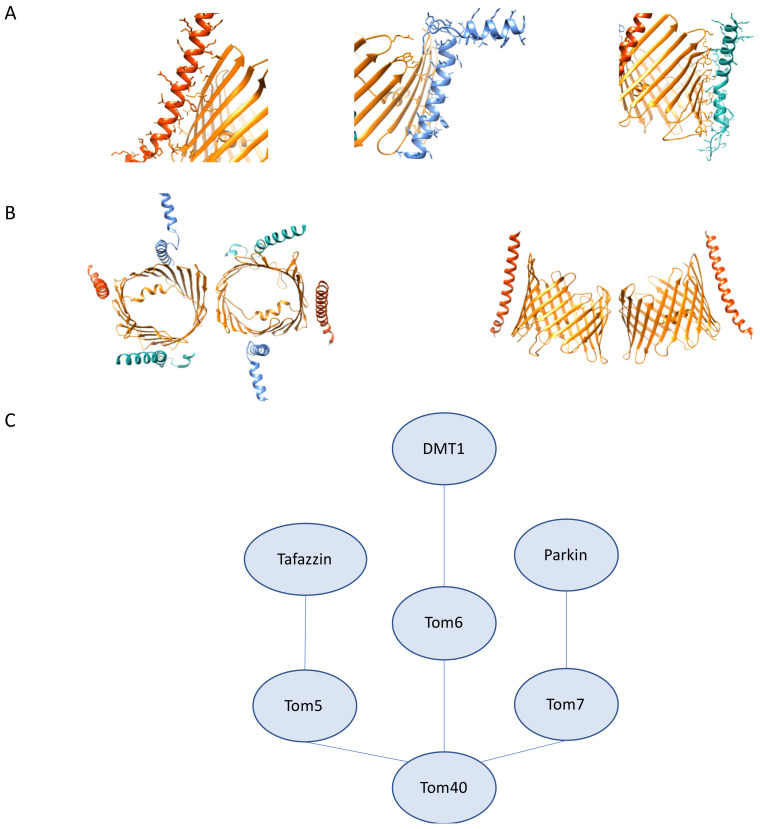
Tom5, Tom6, Tom7 structural features and interactions. (**A**) High resolution structure of human Tom5, 6, 7 (PDB:7ck6). (**left**) Tom5 (vermillion) interactions with Tom40 (orange). (**middle**) Tom6 (light blue) interactions with Tom40 (orange). (**right**) Tom7 (seafoam green) interactions with Tom40 (orange). (**B**) (**left**) Tom5 (vermillion), Tom6 (light blue) and Tom7 (seafoam green) cytosolic view showing arrangement around the β-barrel (**right**). Curvature of Tom5 along Tom40 (orange). (**C**) Tom5, Tom6, Tom7 interactions as described. Distances do not represent interaction strength.

**Figure 7 cells-10-01164-f007:**
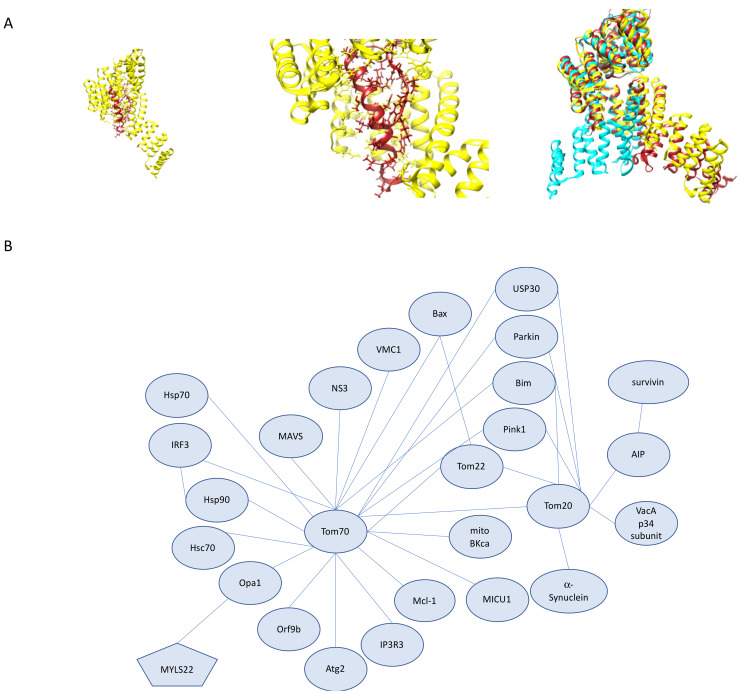
Tom20 and Tom70 Interactions. (**A**) Structure of human Tom70 bound to SARS-CoV-2 protein Orf9b. (**left**) Human Tom70 (yellow) binding pocket with Orf9b (red) bound. (**middle**) Closer view of Orf9b binding pocket with hTom70 (yellow) and Orf9b (red). (**right**) Superimposition of human Tom70 (yellow), yeast Tom70 (cyan) and yeast Tom71 (brick red). Superimposition highlights the rotation of the N-terminal domain between the fungal structures. (**B**) Tom70 and Tom20 interactions as described. Distances do not represent interaction strength.

## Data Availability

No new data were created or analyzed in this study. Data sharing is not applicable to this article.

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
