# Peer review of "A Biochemical and Structural Understanding of TOM Complex Interactions and Implications for Human Health and Disease"

_cells, 2021, doi:10.3390/cells10051164_

Round 1

Reviewer 1 Report

Pitt and Buchanan wrote a very extensive and in-depth review on the rol and function of TOM complex. They describe what is known on the different proteins in different species and their potential evolutionary modulations. Next they extensively describe the potential virus infections and in other human diseases of the different TOM proteins and associations that have been found. 

Here,  are just some small remarks:

  1. Increase the font size in figure 2.
  2. In the figures 3,4 and 6 it is stated that both TOM40, TOM22 and TOM70 interact with e.g. BAX. Please create an overview figure with overlays between the different complexes. Although figure 7 contains such information the others also could decide to in-coorporate this in this figure e.g. by using color coded lines.
  3. Please refer earlier in the text to figure 7, as this is an overview figure and contains all the information on the interactions.
  4. Please refer to the fact that ATG2 is related to autophagy. In figure 7 and therefore it plays an important role in cancer. Same fore e.g. OPA1 and provides an unfavorable prognosis when up-regulated.
  5. Line 159: references are missing
  6. Line 400: To which  type of stress conditions do the authors refer?

Author Response

  1. Font size increased to 9.
  2. Figure 7 has been amended to include color coded lines for ease of interpretation. An additional therapeutic has been included that target TOM complex subunit interaction partners. TOM complex subunits have also been color coded for increased clarity and for consistency from TOM structures
  3. Changed Figure 7 to Figure 2 to provide an earlier introduction into the broad interactome of the TOM complex components.
  4. Used color coded lines to address the role of Atg2 in cancer and MERCS, same for Opa1 as well as included recent citation on therapeutic that targets Opa1.
  5. Updated references to include early acid chain hypothesis works including:
    • Bolliger, L., et al., Acidic receptor domains on both sides of the outer membrane mediate translocation of precursor proteins into yeast mitochondria. EMBO J, 1995. 14(24): p. 6318-26.
    • Dietmeier, K., et al., Tom5 functionally links mitochondrial preprotein receptors to the general import pore. Nature, 1997. 388(6638): p. 195-200.
    • Schatz, G., Just follow the acid chain. Nature, 1997. 388(6638): p. 121-2.
  6. Specified ER stress, and the authors use of brefeldin A to induce ER stress.

Reviewer 2 Report

The review by Pitt and Buchanan describes the current state of the art concerning the Translocase of the Outer Membrane (TOM) complex, the interactions among its component and their link with mitochondrial functionality.

The paper is very interesting and updated; some parts (including figures) report very detailed structural aspects which makes the article less accessible to a wide audience. Conversely, information about development of therapeutics is limited and only sketched, but probably due to incomplete or still poor knowledge about this topic. But overall this is a valuable review.

Major points

1-I found the title not appropriate: the focus is clearly the Translocase of the Outer Membrane (TOM) complex rather than Mitochondrial Outer Membrane Proteins in general.

2-My main objection concerns the “selection” of the interactions/proteins that are reported and discussed in the paper (and depicted in the figures). It is not clear if the authors selected the most interesting proteins, or the proteins with the “higher scores” of interaction, or proteins associated with human diseases...

For instance, as stated by the authors, “Approximately 1100 proteins of the ~ 1500 proteins imported by the TOM complex rely on hTom22 for mitochondrial import”. But the authors described only a handful of such proteins.

Conversely, they also affirmed that “each receptor protein has unique interacting partners involved in disease pathogenesis”. The specificity of the reported interactions should be better clarified.

3-In my opinion, there are few parts which may be slightly improved and concepts which may be added/discussed (linked to point 2):

NDUFS4: why only this respiratory chain complex subunit?

TIM23: please, mention the link between TOM and TIM complexes

TOMM70: please, cite and discuss the paper by Dutta, et al. De novo mutations in TOMM70, a receptor of the mitochondrial import translocase, cause neurological impairment. Hum. Molec. Genet. 29: 1568-1579, 2020.

4- I think that Figure 7 should be amended. The inclusion of the interacting proteins in the different groups is questionable. For instance, Pink and Parkin are related to Neurodegenerative diseases, and to autophagy/mitophagy (rather than to apoptosis); OPA1 and mfn1 are related to mitochondrial morphology (rather than to metabolism)…

“Pentagons represent small molecules”, but there is a single molecule in the figure! It would be better to have additional molecules (potential drugs) in the figure.

Minor points

  1. I think SARS-CoV-2 can be removed from the abstract: the term is quite inflated in this period, but in this review only 10 lines refers to SARS-CoV-2
  2. Line 49. The sentence “In addition to the roles these proteins have when synthesized, the mitochondrial genome plays a critical role in the innate immune response” should be rephrased. One part is about mitochondrial proteins, the other about mtDNA.
  3. Please, replace a- and b- with alpha- (or α) and beta- (or β)
  4. “…domains of Tom70, Tom20, and Tom22, … overlap between the two receptor proteins”

What are the two proteins? Tom70, Tom20, Tom22?

  1. Line 301. Free TOM complex components can be detected by Western blot following blue-native electrophoresis, or through complexome profiling. Is there any such published data supporting the hypothesis of reservoirs of free components?

Author Response

Major points:

  1. Titled updated to address the focus on TOM complex subunit interactions.
  2. Line 80 and Line 86, Line 771: Updated to emphasize focus on human health and disease in the context of selecting interactions, the interactions of TOM complex subunits outside of precursor translocation and how viral proteins can hijack mitochondrial import machinery.
    1. NDUFS4- Line 442-446 provides more detail on the specificity of the Tom40-BAP31 interaction to mitochondrial localization of NDUFS4. Information also included on the specific stress that leads to BAP31-Tom40 complex dissociation and provides more detail on a mechanism by which ER stress has downstream impacts of mitochondrial homeostasis by regulating a particular protein’s import.
    2. TIM- Line 563-568 describes the interaction between TOM complex subunits and TIM23 subunits. Additional detail has been added to provide information on the interactions between the TOM complex and TIM complex however since the focus here is on human disease and function
    3. TOMM70- Lines 312-326 address 4 Tom70 mutants identified in patients with neurological symptoms and also emphasize the importance of structure to understanding molecular implications of mutations.
  3. Figure 7 originally included the roles the proteins play as described in relation to the TOM complex subunit interactions as well as additionally the roles they play in other contexts, and have instead been amended to include color coded lines for ease of interpretation. An additional therapeutic has been included that target TOM complex subunit interaction partners. TOM complex subunits have also been color coded for increased clarity and for consistency from TOM structures

Minor points:

  1. SARS-Cov-2 has been removed from the abstract and changed to a broader sentence on viral proteins and the TOM complex
  2. Sentence rephrased to focus on the function of mtDNA in innate immune response.
    1. Line 54 now states: In addition to encoding for proteins, mitochondrial DNA plays a critical role in the innate immune response
  3. All alpha and betas have been replaced by Greek symbols.
  4. Mentioned the substrate overlap between Tom20 and Tom70, language also changed to specify that we are referring to the loosely associated subunits.

Re: Free TOM complex components: Lines 340-343 Address the current published data that support the hypothesis of TOM complex subunits that are unbound to mature TOM complexes, as well as the dynamic nature of the TOM complex. Language also changed to illustrate that these proteins could still have interaction partners.

Cryo-slicing of Blue Native-MS complexome profiling have identified different assemblies of the TOM complex that includes VDAC-Tom40-Tom22 but this was shown in yeast.